# Design Guidelines for Thermally Driven Micropumps of Different Architectures Based on Target Applications via Kinetic Modeling and Simulations

**DOI:** 10.3390/mi10040249

**Published:** 2019-04-14

**Authors:** Guillermo López Quesada, Giorgos Tatsios, Dimitris Valougeorgis, Marcos Rojas-Cárdenas, Lucien Baldas, Christine Barrot, Stéphane Colin

**Affiliations:** 1Institut Clément Ader (ICA), CNRS, INSA, ISAE-SUPAERO, Mines Albi, UPS, Université de Toulouse, 31400 Toulouse, France; marcos.rojas@insa-toulouse.fr (M.R.-C.); lucien.baldas@insa-toulouse.fr (L.B.); christine.barrot@insa-toulouse.fr (C.B.); stephane.colin@insa-toulouse.fr (S.C.); 2Department of Mechanical Engineering, University of Thessaly, 38334 Volos, Greece; tatsios@mie.uth.gr (G.T.); diva@mie.uth.gr (D.V.)

**Keywords:** Knudsen pump, thermal transpiration, vacuum micropump, rarefied gas flow, kinetic theory, microfabrication, photolithography, microfluidics

## Abstract

The manufacturing process and architecture of three Knudsen type micropumps are discussed and the associated flow performance characteristics are investigated. The proposed fabrication process, based on the deposition of successive dry film photoresist layers with low thermal conductivity, is easy to implement, adaptive to specific applications, cost-effective, and significantly improves thermal management. Three target application designs, requiring high mass flow rates (pump A), high pressure differences (pump B), and relatively high mass flow rates and pressure differences (pump C), are proposed. Computations are performed based on kinetic modeling via the infinite capillary theory, taking into account all foreseen manufacturing and operation constraints. The performance characteristics of the three pump designs in terms of geometry (number of parallel microchannels per stage and number of stages) and inlet pressure are obtained. It is found that pumps A and B operate more efficiently at pressures higher than 5 kPa and lower than 20 kPa, respectively, while the optimum operation range of pump C is at inlet pressures between 1 kPa and 20 kPa. In all cases, it is advisable to have the maximum number of stages as well as of parallel microchannels per stage that can be technologically realized.

## 1. Introduction

The rapid development of the semiconductor industry has been followed, in the last decades, by huge progress in microfabrication processes, providing a large number of micro-electro-mechanical systems (MEMS). Some of these systems, such as lab-on-a-chip (LOC) or micro total analysis systems (µTAS) for gas sensing, analyzing and separation, as well as for drug delivery, require an external pumping system to move the gas samples through the various stages of the device [1]. Additionally, radio frequency switches, vacuum tubes, and other components that depend on electron or ion optics require a stable vacuum environment for proper operation. Since merely sealing these devices is not sufficient to guarantee long-term operation free of leakages and outgassing, miniaturized vacuum pumping components are needed to maintain proper functionality [2].

Thermal transpiration (or thermal creep) has been known for more than 100 years, since the first studies of Maxwell [3], Reynolds [4], and Knudsen [5,6]; however, functional pumping prototypes based on this phenomenon have been developed only during the last 20 years with the massive development and adoption of microfabrication techniques. The Knudsen pump, which is one of the devices exploiting the thermal transpiration phenomenon, is able to generate a macroscopic gas flow by applying exclusively a tangential temperature gradient along a surface without any moving parts or external pressure gradient [7,8,9,10,11,12]. The induced thermal transpiration flow is useful for technological purposes, provided that the flow is through microscale channels, since the phenomenon is intensified as the characteristic length of the system is decreased. Since the Knudsen pump only requires a temperature gradient for its operation, its architecture is quite simple and it does not require any moving parts, which provides high reliability and avoids any maintenance. Its advanced compactness allows low power consumption. Furthermore, since the direction of the flow can be reversed by inverting the thermal gradient in the microchannels, the Knudsen pump could provide significant benefits for sampling and separation devices [13,14]. The classic architecture of the Knudsen pump consists of a series of wide channels (or reservoirs) connected by microchannels [15] or a porous medium [16]. In order to avoid large temperature gradients, and since the pumping effect generated by a single stage with a moderate temperature gradient is not adequately strong, a multistage system is commonly applied, with a periodic temperature variation in each stage. The thermal transpiration flow produced in the microchannels or the porous medium is much larger than the counter one in the wide channels and therefore, a net pumping effect, which is expected to be increased with the number of stages, is obtained. This architecture presents various difficulties related to microfabrication and local thermal gradients control and therefore, the progress in the field has been limited.

Using advanced microfabrication techniques, a few functional prototypes of Knudsen pump have been recently developed. The first microfabricated single-stage Knudsen pumps were reported in [17], where a nanoporous aerogel was used as a transpiration membrane, and in [15], where the pump was formed of constant cross section microchannels. Some works have integrated Knudsen pumps in fully integrated MEMS and several multistage Knudsen pumps with various operation characteristics have been developed [16,18,19,20], demonstrating low power consumption (less than 1 W) [19] and long operating times (more than 11750 h of continuous operation) [16]. In order to further decrease the power consumption of these devices, the Knudsen pump can also be powered by passive heat recovery from other processes. Finally, a fully electronic micro gas chromatography system integrated with all its fluidic components has been recently achieved, including a bi-directional Knudsen pump with a mesoporous mixed cellulose‒ester membrane sandwiched between two glass dies to provide the parallel flow channels, significantly reducing the fabrication complexity and cost [21,22].

In the present work, specific design guidelines for manufacturing Knudsen pumps by means of a low thermal conductivity bulk material and an innovative microfabrication process are described in order to further improve manufacturing and operational issues concerning flexibility, adaptability, thermal management and overall performance. Three Knudsen pump designs, depending upon the target application, are proposed. The associated numerical investigation is performed to demonstrate the versatility of the proposed designs.

## 2. Proposed Pump Designs, Manufacturing Materials, and Fabrication Process

Concerning the performance characteristics of Knudsen pumps and their corresponding targeted applications, there are, in general, three alternative designs, named below as designs A, B, and C. The first one (design A) targets applications such as micro-gas chromatography, where the needed mass flow rate is high while the corresponding pressure difference is small. This goal can be reached by designing a large number of channels in parallel to increase the overall mass flow rate. As there is no need to generate a strong pressure difference, this first configuration requires only one stage, which simplifies the design. The second design (B) targets applications such as vacuum maintenance in low-power devices where the needed pressure difference is high, while the corresponding mass flow rate is not so relevant. It requires a large number of stages to increase the pressure difference, which is limited for each stage by the available temperature difference. In order to increase compactness and decrease power consumption, each stage is made of a single channel, as a small mass flow rate is acceptable for this kind of application. Finally, the third design (C) is less specific and targets applications where both relatively high mass flow rates and pressure differences are needed. The third design may be employed in MEMS, as well as in other applications potentially substituting conventional pumps.

In this work, all three pump designs (Figure 1) are considered to address the corresponding performance characteristics:Pump A consists of an array of multiple parallel narrow microchannels in one single pumping stage to achieve high mass flow rate ($\dot{m}$) performance. The layout area is a×a with n parallel narrow microchannels of diameter d and length L.Pump B consists of a multistage system where each stage is formed by one single narrow pumping microchannel followed by one wide channel (where the reduced counter thermal transpiration flow will appear) to achieve high pressure difference (ΔP) performance. The layout area is (W×W)+(b×b). The diameter of the narrow and wide channels are d and D, respectively, and the length of both channels is L.Pump C combines the two previous designs. More specifically, it consists of a multistage system and each stage is formed by an array of n parallel narrow pumping microchannels, followed by one wide channel where the reduced counter thermal transpiration flow will appear. This design provides high ΔP and $\dot{m}$ performances, due to the multi-stage cascade system and to the multiple narrow microchannels per stage, respectively. The layout area is (W×W)+(c×c). The diameter of the narrow and wide channels are d and D, respectively, while the length of all channels is L.

Simulations of pumps A, B, and C have been performed via kinetic modeling, described in Section 3 and the performance characteristics are analyzed, in terms of the parameters affecting the flow, in Section 4.

Most high-precision microfabrication processes have been initially based on silicon micromachining as the semiconductor industry has been leading the market. However, silicon displays a high thermal conductivity that increases the heat transfer through the solid between the hot and cold reservoirs and results in a high power consumption, while increasing the difficulty of adequately controlling the temperature gradient along the microchannels. On the contrary, by using bulk materials with lower thermal conductivity, such as glass or polymers, the thermal management of the device is simplified and the performance can be improved by maintaining higher temperature gradients in simpler structures. Based on all the above information, we propose using polymer instead of silicon as the bulk material. On the other hand, the substrate used for connecting the heating and cooling elements, respectively, to the hot and cold regions at the top and the bottom of each stage, can be made of silicon to improve the temperature uniformity in these hot and cold regions.

The proposed Knudsen pump designs can be realized by the following innovative fabrication process. Instead of etching, the technique of the deposition of dry film (DF) photoresist layers [23] can be implemented in a cost- and time-effective way. The proposed DF photoresist approach is based on a negative epoxy, which is a low-cost material that can be combined with standard photolithography procedures and multilevel laminating by rolling the DF layers with a specific pressure and temperature. Each new layer can then be stacked on the previous one without damaging the patterns of other layers, which leads to the creation of 3D structures, as shown in Figure 2, and allows for fabricating multistage Knudsen pumps. The fabrication process is summarized as follows: (i) the DF photoresist layer (uncrosslinked DF) is laminated onto a planar substrate (glass or silicon wafer); (ii) following standard photolithography procedures, the DF layer is exposed to UV light and baked; (iii) the DF layer is developed (reticulated DF) in a bath of cyclohexanone that removes the material of the non-exposed areas during the photolithography process. This process is repeated for each of the successive layers, enabling the production of different patterns [23]. The alignment of each layer has been reported with a deviation of less than 1 µm. Also, since the typical thicknesses of the commercialized DF layers go from 5 to 100 µm, various layers can be stacked for a specific pattern, to increase and adjust the thickness of that particular pattern (i.e., the length of the channels).

Furthermore, the proposed pump designs are to be fabricated in such a way that successive stages are in the same block and the only connections of the pump are reduced to one inlet and one outlet. The possible sources of leakage, which are common in gas microsystems due to numerous connections, are thus drastically reduced. In addition, manufacturing the microchannels across the thickness of each layer keeps the hot and cold regions of the Knudsen pump spatially separate. Consequently, it simplifies the thermal management and allows easy bi-directional pumping. The proposed manufacturing process is quite flexible in terms of geometry, and Knudsen pump prototypes with various shapes, sizes and lengths can easily be produced. Actually, this approach with the channels through the thickness of the substrate and the hot and cold regions spatially separated, has already been employed for Knudsen pumps with high flow (with a design similar to that of pump A) in [20,21,22] and is generalized here in all three designs.

## 3. Kinetic Modeling

The flow configurations in the three proposed pump designs are modeled in order to obtain the expected performance for each case. Since the flow has a wide range of Knudsen number, kinetic modeling is introduced. Furthermore, since the length of the narrow microchannels is always much longer than the radius, the flow may be considered fully developed, i.e., the pressure varies only in the axial direction and remains constant in each cross section of the capillary. Thus, flow modeling is based on the infinite capillary theory, which is well known and established for pressure- and temperature-driven rarefied gas flows [24,25,26,27,28,29,30]. Additionally, in cases where the fully developed assumptions are not met (i.e., in the wide channels of diameter *D* subject to thermally driven back flow), the end effect correction is accordingly introduced [31,32]. The correction is introduced only in the pressure and not in the corresponding temperature-driven flow, since the mass flow rate in the former one is about one order of magnitude larger than in the latter.

Consider the fully developed thermal transpiration flow of a monatomic variable hard sphere molecule through a circular channel with length L and radius R (with R<<L) that connects two reservoirs maintained at different temperatures $T_{C}$ and $T_{H}$, with $T_{C}<T_{H}$. In addition to the temperature-driven flow, a pressure-driven flow is generated. Similar to the analysis performed in [30] for thermally driven flow through tapered channels, the net mass flow rate $\dot{m}$ may be computed based on the differential equation
(1)$$\frac{dP}{dz}=-\frac{\dot{m}\upsilon_{0}\left(z\right)}{\pi R^{3}G_{P}\left(\delta \right)}+\frac{G_{T}\left(\delta \right)}{G_{P}\left(\delta \right)}\frac{P\left(z\right)}{T\left(z\right)}\frac{dT}{dz},$$
subject to the given pressures P(0) and P(L) at the channel inlet and outlet, respectively. Here, z∈[0,L] is the coordinate along which the flow is directed, $\upsilon_{0}\left(z\right)=\sqrt{2R_{g}T(z)}$ is the probable molecular speed, $R_{g}$ is the specific gas constant, T(z) is the imposed linear temperature distribution along the channel wall, and P(z) is the unknown pressure distribution and it is part of the solution. Additionally, $G_{P}\left(\delta \right)$ and $G_{T}\left(\delta \right)$ are the dimensionless flow rates, also known as kinetic coefficients [24,25,27,33] for the pressure- and temperature-driven flows, respectively. They depend on the local gas rarefaction parameter, defined as
(2)$$\delta \left(z\right)=\frac{P\left(z\right)R}{\mu \left(z\right)\upsilon_{0}\left(z\right)}\;,$$
where μ(z) is the local dynamic viscosity. In the present work, the kinetic coefficients $G_{P}\left(\delta \right)$ and $G_{T}\left(\delta \right)$ are retrieved from a kinetic database, which has been developed based on the linearized Shakhov model subject to pure diffuse boundary conditions in the range δ∈[0,50]. Additionally, when δ>50, the analytical slip solution is used. For completeness, tabulated results and the analytical slip solution of the kinetic coefficients versus gas rarefaction parameter are presented in Appendix A.

Equation (1) is solved for $\dot{m}$ and P(z) as follows: an initial value for the mass flow rate $\dot{m}$ is assumed and Equation (1) is integrated with initial condition $P_{in}=P\left(0\right)$ along z∈[0,L]. At each integration step along z, the values of $G_{P}$ and $G_{T}$ are updated based on the local δ(z). The computed outlet pressure is compared to the specified $P_{out}=P\left(L\right)$ and then the mass flow rate is corrected depending on the difference between the computed and specified outlet pressures. This procedure is repeated until $\dot{m}$ and the corresponding P(z) converge to yield the specified P(L).

The above description refers to a single channel and may be applied in a straightforward manner to pumps A, B, and C, provided that the carrier gas, the pump geometry, the temperature variation along the channels, and the inlet and outlet pressures are all specified. In each case the computed mass flow rate $\dot{m}$ with the associated pressure difference $\Delta P=P_{out}-P_{in}$ between the inlet and the outlet fully characterize the pump performance. It is noted that a very close kinetic type analysis of pump B, based on the ellipsoidal‒statistical (ES) model, and the one presented here, based on the Shakhov model, has been performed in [10].

## 4. Results and Discussion

The performance characteristics of the three proposed pump designs are computationally investigated, taking into account manufacturing and operational constraints. Proper thermal management is of major importance. Previous experimental works on thermal transpiration flows through capillaries [34,35], supplemented by recent typical heat transfer simulations in film layers, clearly indicate that it is possible to provide temperature differences $\Delta T=T_{H}-T_{C}$ on the order of 100 K within a channel of length L=300 μm by integrating active heating and cooling. In addition, via the proposed microfabrication process, microchannels with diameters ranging from 100 down to 5 μm are attainable with L=300 μm by deposition of successive layers of dry film photoresist. This length corresponds to the thickness of each layer and is kept small in order to minimize the pump volume. In a later stage, smaller diameters may be fabricated by further reducing the channel length, i.e., the film layer thickness. 

Based on the above, simulations for pump designs A, B, and C consisting of narrow and wide microchannels with diameters d=50,20,10,5 μm and D=100 μm, respectively, always keeping the length L=300 μm, have been performed. As pointed out before, the pumping effect is produced by the thermal transpiration flow in the narrow microchannels, while the counter thermal transpiration flow in the wide microchannels is much smaller due to the larger diameter of the channel and, therefore, there is a net flow in the pumping direction. The temperature difference is set to ΔT=100 K, maintaining the cold and hot temperatures at $T_{C}=300$ K and $T_{H}=400$ K, respectively. The gas considered here is argon with a specific constant $R_{g}=208.1{\;J\;\mathrm{kg}}^{-1}K^{-1}$ and a dynamic viscosity varying with temperature according to an inverse power-law model, which is consistent with the variable hard sphere molecule hypothesis [11]: $\mu \left(z\right)=\mu_{0}{\left(T\left(z\right)/T_{0}\right)}^{0.81}$ with a viscosity $\mu_{0}=2.211\times 10^{-5}\mathrm{Pa}\;s$ at the reference temperature $T_{0}=273\;K$.

Furthermore, taking into account the space needed between the channels from the fabrication point of view, it can be estimated that the minimum total square area allowing an array of at most n=400 microchannels is 200×200 μm^2^. A schematic view of the corresponding layouts to be examined is shown in Figure 3. As the microchannel diameter is reduced, the number *n* of parallel microchannels in the layout is increased, keeping the same area ratio between the flow and the layout cross section areas. In this way, the comparison of the performance characteristics of the different layouts always involves the same cross section flow area. Additional details of the layout geometry are provided in Table 1. 

The computational investigation includes pumps A, B, and C and is presented in Section 4.1, Section 4.2, and Section 4.3, in terms of the inlet pressure $P_{in}\in \left[0.1-10^{5}\right]$ kPa, the diameter d of the narrow microchannels, and the number N of stages.

### 4.1. Pump A: One Pumping Stage with Multiple Parallel Microchannels

Pump A consists of a single pumping stage with *n* parallel microchannels, mainly targeting high mass flow rates $\dot{m}$ and small pressure differences. It is obvious that, as the channel diameter is reduced, the flow rate in a single channel, $\dot{m}$, is also reduced. This mass flow rate loss is partly compensated for, since as the diameter is decreased more channels are packed in parallel, keeping the same cross section area. The total mass flow rate will be equal to that of a single channel multiplied by the number *n* of parallel channels. On the contrary, the total pressure difference will be equal to that of each channel, which increases as the channel diameter decreases. Therefore, first, a comparison is performed between the performances of the one-stage pumping for the different layout geometries by varying the narrow diameter and the number of channels, while preserving the same total surface, as shown in Figure 3 and Table 1.

The investigation is performed by computing the total maximum mass flow rate ${\dot{m}}_{n}$ through the *n* parallel channels corresponding to zero pressure difference, i.e., to a pump with the same pressure at both ends, as well as the maximum pressure difference $\Delta P_{n}$ corresponding to zero net mass flow rate (closed system). For n=4,25,100, and 400, ${\dot{m}}_{n}$ and $\Delta P_{n}$ are compared with the reference values ${\dot{m}}_{1}$ and $\Delta P_{1}$ respectively, corresponding to the case of a single channel, n=1. The channels diameters associated with n=1,4,25,100, and 400 are d=100,50,20,10, and 5 µm, respectively. The ratios ${\dot{m}}_{1}/{\dot{m}}_{n}$ and $\Delta P_{n}/\Delta P_{1}$ are computed to estimate the mass flow rate decrease and the pressure difference increase when the number of channels is increased and their diameter is decreased. The results are presented in Figure 4 as a function of the inlet pressure $P_{in}$.

As expected, as the diameter is decreased and the number of microchannels is increased, both ${\dot{m}}_{1}/{\dot{m}}_{n}$ and $\Delta P_{n}/\Delta P_{1}$ are monotonically increased, meaning that the total flow rate through the parallel channels decreases and the pressure difference increases. In terms of the inlet pressure, it is seen that at a low inlet pressure the ratios of the total mass flow rates ${\dot{m}}_{1}/{\dot{m}}_{n}$ are larger than the corresponding ratios of pressure differences $\Delta P_{n}/\Delta P_{1}$, while at moderate and high inlet pressures, the situation is reversed and the ratios ${\dot{m}}_{1}/{\dot{m}}_{n}$ become smaller than the corresponding $\Delta P_{n}/\Delta P_{1}$. This behavior clearly implies that, in the microchannels, the relative pressure difference increase compared to the corresponding total mass flow rate decrease is less significant in highly rarefied conditions, while in less rarefied conditions closer to the slip and hydrodynamic limit, it becomes more significant. Furthermore, the solid red circles indicate the inlet pressure values where the two ratios are equal. Then, at the left side of these points, the ratios of the mass flow rates ${\dot{m}}_{1}/{\dot{m}}_{n}$ are larger than the corresponding ratios of pressure differences $\Delta P_{n}/\Delta P_{1}$, while at the right side $\Delta P_{n}/\Delta P_{1}$ are larger than ${\dot{m}}_{1}/{\dot{m}}_{n}$. For $P_{in}\geq 5$ kPa, whatever the number *n* of parallel microchannels, the relative variation in pressure difference is much more significant than the relative variation in the total mass flow rate through the combined parallel channels.

To have a clear view of the performance characteristics of the one-stage Knudsen pump A in absolute quantities (not in relative ones, as previously), the pressure difference and the mass flow rate through a single channel are plotted in Figure 5, for microchannels with d=5,10,20 μm, as a function of the inlet pressure $P_{in}$. In all cases, as the inlet pressure is increased, the pressure difference is initially rapidly increased, reaching its peak value, and then is slowly decreased. The peak values $\Delta P_{peak}$ occur, depending upon the diameter d, at about $P_{in}=4-10$ kPa, corresponding to values of the gas rarefaction parameter δ=3−4, i.e., in the transition regime. This behavior has also been observed and reported in experimental studies and is independent of the working gas, the channel characteristic length, and the temperature difference [34]. Depending on the inlet pressure, the generated pressure difference ΔP varies from 10 to 420 Pa and, as expected, the obtained pressure difference is increased as the diameter is reduced with the maximum ΔP=420,200,100 Pa for d=5,10,20 μm, respectively. Regarding the mass flow rate, it is rapidly increased as the inlet pressure is initially increased and then it keeps increasing at a much slower pace until it becomes constant in the hydrodynamic regime. For $P_{in}\geq 20$ kPa, the mass flow rate is almost stabilized and reaches values of about $0.5\times 10^{-12}$, $2.1\times 10^{-12}$ and $8.5\times 10^{-12}$ kg/s for d=5,10,20 μm, respectively. Multiplying these values with the corresponding number of microchannels, n=400, 100 and 25, results in mass flow rates higher than $10^{-10}$ kg/s. These results, properly scaled, are in good agreement with the corresponding ones in [20], where a design similar to pump A was realized.

Overall, it may be stated that, for pump A, it is preferable to have a wider channel to increase the total mass flow rate rather than a large number of parallel microchannels with smaller diameters. However, since reducing the channels diameter results in huge gains of ΔP with only small reductions of ${\dot{m}}_{n}$, as shown in Figure 4, depending on the application, it might be advisable to reduce the diameter to boost ΔP while only slightly reducing the overall mass flow rate ${\dot{m}}_{n}$. Finally, it is preferable to operate the pump with moderate and high inlet pressures ($P_{in}>5$ kPa), where the mass flow rate starts to stabilize at large values.

### 4.2. Pump B: Multistage Pumping with One Narrow and One Large Channel Per Stage

Pump B is a multistage system, where each stage consists of one single pumping narrow microchannel with d=50,20,10,5 μm, followed by one wide channel with D=100 μm, targeting high pressure differences and small mass flow rates (as there is only one microchannel per stage).

The maximum pressure difference corresponding to zero net mass flow rate (closed system) at various inlet pressures $P_{in}=1,5,10,20,50,100$ kPa are provided in Figure 6a,b versus the number of stages with N≤1000 and N≤100, respectively. The considered narrow and wide microchannels at each stage have diameters d=10 μm and D=100 μm, respectively. As seen in Figure 6a, the pressure difference, ΔP, increases with the number of stages in a qualitatively similar manner for all inlet pressures, except for the lowest inlet pressure $P_{in}=1$ kPa and N≤100. It is also seen that for a small number of stages the pressure difference is rapidly increased and then it keeps increasing, but at a slower pace that gradually decreases as the number of stages increases. This is due to the fact that, for all the inlet pressures shown, except $P_{in}=1$ kPa, the pumps, independently of N, always operate in the decreasing region of ΔP in terms of $P_{in}$ (see Figure 5a). In all these cases, each time a stage is added its contribution to the overall ΔP is slightly reduced compared to the previous one, because the inlet pressure of the stage is increased. Therefore, the rate at which ΔP is increased is slowly reduced with increasing number of stages. In the specific case of $P_{in}=1$ kPa the pump starts to operate in the increasing region of ΔP in terms of $P_{in}$ (see Figure 5a). Consequently, starting at $P_{in}=1$ kPa, every added stage, compared to the previous one, contributes with a larger ΔP until the pressure of $P_{in}=5$ kPa, corresponding approximately to the peak value of the pressure difference $\Delta P_{peak}$, is reached. Thus, a more detailed view for N≤100 is shown in Figure 6b. It is clearly seen that the pressure difference with respect to the number of stages for $P_{in}=1$ kPa is qualitatively different compared to all other $P_{in}$ values and is increased more rapidly. Of course, once a sufficient number of stages has been added and the inlet pressure for the next stage becomes higher than $P_{in}=5$ kPa, then the pump operates in the decreasing region of ΔP in terms of $P_{in}$ and it behaves qualitatively similar to all others. Overall, it is observed that for a multistage pump with N≥100 the pressure difference generated is very significant and may be of the same order as the inlet pressure or even several times higher, when $P_{in}\leq 20$ kPa. It is noted that the present results are in excellent qualitative agreement with the corresponding ones reported in [10]. Furthermore, running the present code for some of the geometrical and operational parameters in [10], it has been found that the deviation in the numerical results is small (about 10%) and is due to the corresponding deviation between the kinetic coefficients obtained by the ES and Shakhov models.

Although pump B targets high ΔP and the mass flow rate is not of major importance, it is interesting to observe its variation versus the number N of stages. In Figure 7, the maximum mass flow rate corresponding to zero pressure difference (i.e., a pump with the same pressure at both ends) is shown for the same parameters as in Figure 6a. The mass flow rate is of the order of $10^{-12}$ or $10^{-13}$ kg/s and it is low since only one narrow channel is considered per stage. As expected, $\dot{m}$ is decreased as $P_{in}$ is decreased, but more importantly, $\dot{m}$ is kept constant as the number of stages is increased due to the fact that all stages have the same geometry and the same inlet and outlet pressures. It may be stated that for Knudsen pumps based on the architecture of pump B, it is desirable to add as many stages as possible and to operate the pump in low and moderate $P_{in}$, since ΔP grows with N, while the maximum $\dot{m}$ remains constant. The corresponding results for other narrow microchannel diameters have a similar qualitative behavior.

The evolution of the pressure inside a system vacuumed with pump B connected at its inlet, while the outlet is at the atmospheric pressure ($P_{out}=100$ kPa), is analyzed in Figure 8, showing the influence of the number of stages and of the narrow channels diameter d=5,10,20 μm, with the diameter of the wide channel being always D=100 μm. Three pressure drop regions can clearly be identified: at high pressures (green region), the pressure is reduced very slowly; at intermediate pressures (red region), the pressure is rapidly reduced; and finally, at low pressures (blue region), the pressure keeps reducing at a slower pace. This behavior can be understood by observing the corresponding results in Figure 5, where the pressure difference is very small at high inlet pressures, then becomes quite large at intermediate inlet pressures, where the maximum pressure difference is reported (red symbols in Figure 8) and, finally, becomes small again at low inlet pressures. These regions of Figure 5 correspond to the green, red, and blue regions in Figure 8. As expected, the required number of stages to reach the final low inlet pressure is increased by increasing the diameter of the narrow microchannels. These plots confirm that the optimal operating pressure range of the pumps B presented is, as pointed above, in the red zone (Figure 8), i.e., at moderate and low inlet pressures corresponding to values of the gas rarefaction parameter close to δ=3−4 where $\Delta P_{peak}$ is found (Figure 5).

### 4.3. Pump C: Multistage Pumping with Multiple Parallel Microchannels Per Stage

Pump C is a multistage system, where each stage consists of n parallel pumping narrow microchannels, followed by one wide channel, targeting both high pressure differences and mass flow rates. The total pressure difference is obtained by adding the pressure difference of each stage (shown in Figure 6), while the total mass flow rate ${\dot{m}}_{n}$ is calculated by multiplying the mass flow rate of a single microchannel multistage system (shown in Figure 7) times the number *n* of parallel microchannels.

The performance curves of pump C, showing the variation of pressure difference ΔP versus the total mass flow rate ${\dot{m}}_{n}$, are plotted in terms of the number of stages N in Figure 9 for N=1,5,10,20 and in Figure 10 for N=40,100,200,500,1000, with inlet pressures $P_{in}=1,5,20,100$ kPa. They are presented in different figures for clarity purposes, since going from N=1 up to N=1000 the pressure difference is increased about two orders of magnitude. The narrow microchannels and wide channel at each stage have diameters d=10 μm and D=100 μm, respectively, while the corresponding characteristic curves for other narrow microchannels diameters have a similar qualitative behavior. The values of ΔP for ${\dot{m}}_{n}=0$ and of ${\dot{m}}_{n}$ for ΔP=0, denoted by $\Delta P_{\max}$ and ${\dot{m}}_{n,\max}$, respectively, are considered as the limiting cases corresponding to a closed system with no net mass flow rate and to an open system with the same pressure at both ends. The characteristic performance curves of the pump are defined by these two limiting values and all flow scenarios in between with $0<\dot{m}<{\dot{m}}_{n,\max}$ and $0<\Delta P<\Delta P_{\max}$. As expected, in all the cases the characteristic performance curves exhibit a pressure difference decrease as the mass flow rate is increased. Since the maximum pressure difference is increased with the number of stages, while the maximum mass flow rate is constant independently of the number of stages (see the discussion in Figure 6 and Figure 7), the mean slope of the characteristic curves is increased with the number of stages. For a specified mass flow rate, the developed pressure difference is increased as the number of stages is increased; similarly, for a specified pressure difference, the produced mass flow rate is increased with the number of stages. The largest pressure differences are observed at $P_{in}=1$ kPa and $P_{in}=5$ kPa for N≥100 and N<100, respectively, as shown in Figure 6. On the contrary, as expected, the mass flow rates are monotonically increased with $P_{in}$, with a gain of a factor of 5 when the inlet pressure is increased by a factor of 100. Furthermore, it is seen that the performance curves in Figure 9 are close to linear. This is justified, since in pumps with a small number of stages, the output pressure is relatively close to the inlet pressure and the corresponding variation of δ and $G_{P}\left(\delta \right)$ along the stages of the pump is small, resulting in almost constant terms in Equation (1). However, as N is increased, the outlet pressure is also increased, resulting in a more significant variation of $G_{P}\left(\delta \right)$ along the pump stages and therefore, the performance curves exhibit a nonlinear behavior, as the terms in Equation (1) develop a larger variation. This behavior starts to appear from N=20 in Figure 9 and becomes more evident as N is further increased in Figure 10. Another interesting issue is that, as $P_{in}$ is decreased and N is increased, the pressure difference ΔP decreases very slowly as the mass flow rate increases and then, when the mass flow rate approaches its maximal value ${\dot{m}}_{n,\max}$, the pressure difference rapidly decreases. This behavior, clearly shown in Figure 10, with $P_{in}=1$ kPa, is beneficial for pumps operating at low inlet pressures.

In order to maximize the performance of pump C, it is preferable to operate at low and moderate inlet pressures in the wide range of $P_{in}\in \left[1-20\right]$ kPa, with as many parallel microchannels per stage and number of stages as possible, to take advantage of the flattening of the performance curves but not too close to the maximum mass flow to avoid the abrupt decrease of the pressure difference.

## 5. Concluding Remarks

The manufacturing process and the structure architecture of three thermally-driven micropump designs targeting specific applications have been proposed and the associated performances have been examined. The microfabrication process includes the development of low thermal conductivity dry film layers via lamination on glass, exposition to UV lighting and baking via photolithography procedures and finally, removal of the non-exposed material in cyclohexanone bath. The three pump designs include single-stage pumping through many parallel microchannels targeting high mass flow rates (pump A), multistage pumping with each stage composed of a single narrow microchannel and a wide channel targeting high pressure differences (pump B) and a combination of the former two configurations targeting both high mass flow rates and pressure differences (pump C). Modeling is based on kinetic theory via the infinite capillary approach.

The proposed microfabrication process is fast, easily realized, adaptive to specific applications and cost effective. The thermal management of the pumps is significantly improved because of the involved low thermal conductivity materials and the separation of the hot and cold areas in two different surfaces. The computational investigation has taken into account all foreseen manufacturing and operational constraints, and the optimum performance conditions as a function of the inlet pressure and pump geometry (number of parallel microchannels per stage and number of stages) have been identified. Micropumps based on the architecture of pumps A and B operate more efficiently at inlet pressures higher than 5 kPa and lower than 20 kPa, respectively. In addition, it is advisable to manufacture pump A with as many parallel microchannels as possible and pump B with as many stages as possible. Pump C should be built with both a high number of parallel microchannels per stage and a high number of stages. Furthermore, their optimum operation range is found to be at inlet pressures between 1 kPa and 20 kPa and a working regime not too close to the limiting maximum mass flow rate.

Using the proposed materials and fabrication process, Knudsen pump prototypes based on pump C architecture, with 2000 stages, 100 microchannels of diameter d=10 μm, and one wide channel of diameter D=100 μm per stage, employing a total layout surface of 160 mm^2^, are presently under construction. Since the theoretical pressure differences cover several orders of magnitude, the device could decrease the pressure of a system from atmospheric pressure $P_{in}=100$ kPa down to 1 or 2 Pa, with associated mass flow rates higher than $10^{-10}$ kg/s. These theoretical performances shall be corrected to deal with specific operational constraints such as leakages, issues in thermal management of the device, etc.

## Figures and Tables

**Figure 1 micromachines-10-00249-f001:**
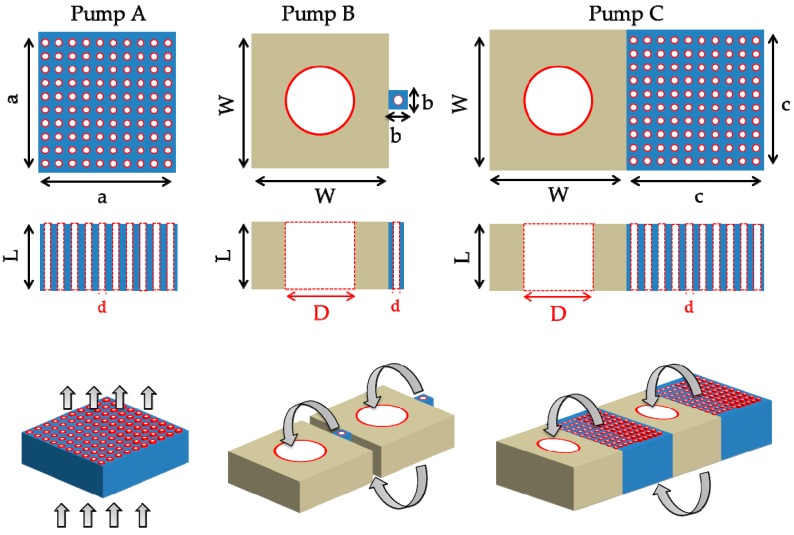
Representative view of single-stage pump A, and of two consecutives stages of multistage pumps B and C. Gray arrows denote the pumping flow direction, the hot and cold regions being at the top and the bottom of each stage, respectively.

**Figure 2 micromachines-10-00249-f002:**
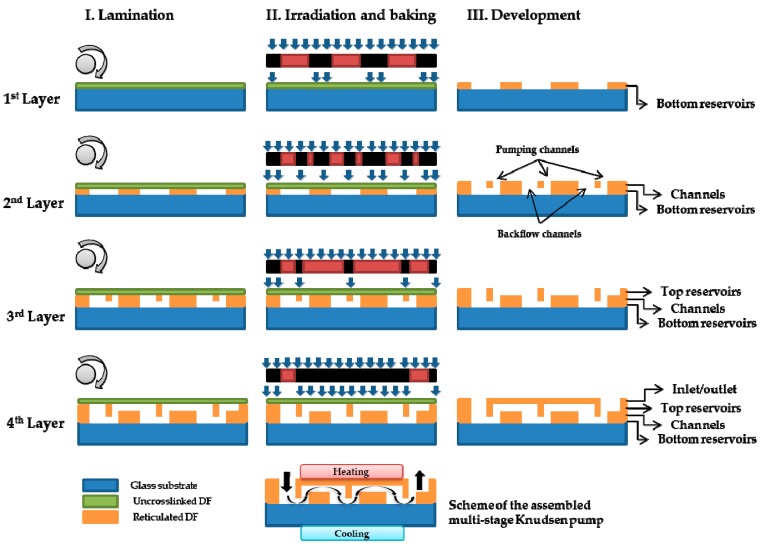
Schematic of fabrication process with the superposition of dry film (DF) photoresist layers and the use of lamination (grey cylinders) and lithography techniques. Typical thicknesses of the DF layers are 5, 25, 50, and 100 µm, while the thickness of the glass wafer is 500 µm. In the cooled reservoirs the glass substrate can be replaced with a silicon substrate to improve temperature uniformity.

**Figure 3 micromachines-10-00249-f003:**
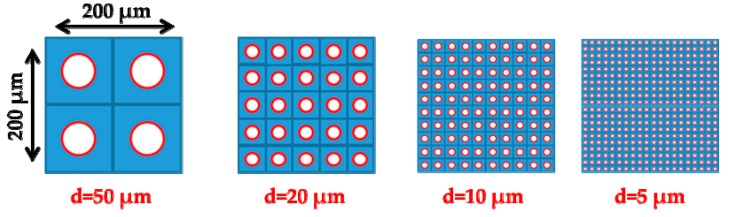
View of layouts with increasing the number *n* of microchannels and decreasing the channel diameter *d*, keeping the same ratio between the channels and the overall cross sections.

**Figure 4 micromachines-10-00249-f004:**
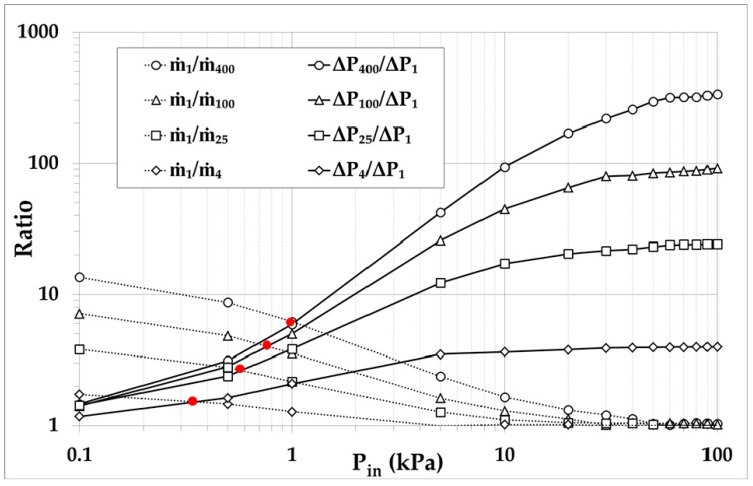
Ratios of total maximum mass flow rates ${\dot{m}}_{1}/{\dot{m}}_{n}$ and associated maximum pressure differences $\Delta P_{n}/\Delta P_{1}$ for various one-stage layouts (numbers of microchannels n=4,25,100,400 and corresponding diameters d=50,20,10,5 μm) compared to the reference layout (n=1, d=100 μm).

**Figure 5 micromachines-10-00249-f005:**
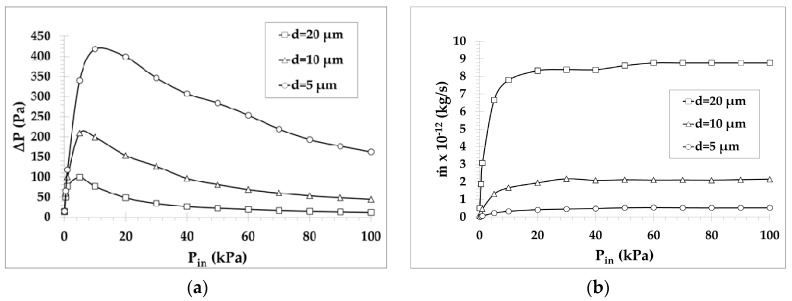
(**a**) Pressure difference and (**b**) mass flow rate versus inlet pressure for the one-stage pump A with d=5,10,20 μm (mass flow rates are given for a single channel of the pump). The rarefaction parameter is ranging from 0.03 to 130, and it increases with the inlet pressure and the diameter, providing the maximum pressure difference at δ=3−4, in the transition regime.

**Figure 6 micromachines-10-00249-f006:**
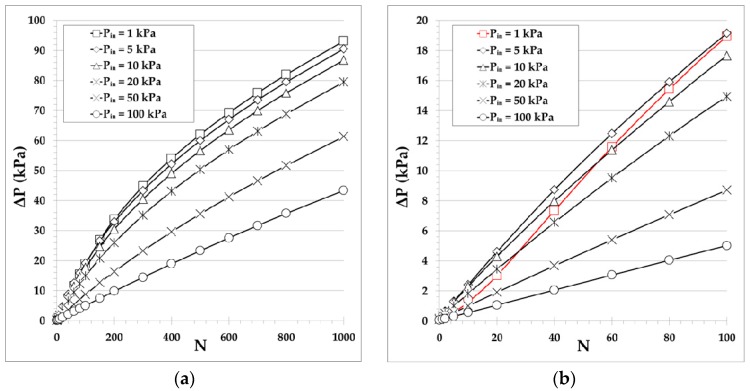
Maximum pressure difference (corresponding to zero net mass flow rate) of pump B with single narrow and wide microchannels of diameters d=10 μm and D=100 μm, respectively, in each stage, at various inlet pressures, versus the number of stages with (**a**) N≤1000 and (**b**) N≤100. The range of the rarefaction parameter in the narrow channel is (**a**) δ≈0.6−56 and (**b**) δ≈0.6−12 for $P_{in}=1$ kPa, and (**a**) δ≈60−86 and (**b**) δ≈60−62 for $P_{in}=100$ kPa. Always, δ is increasing with the inlet pressure and the number of stages.

**Figure 7 micromachines-10-00249-f007:**
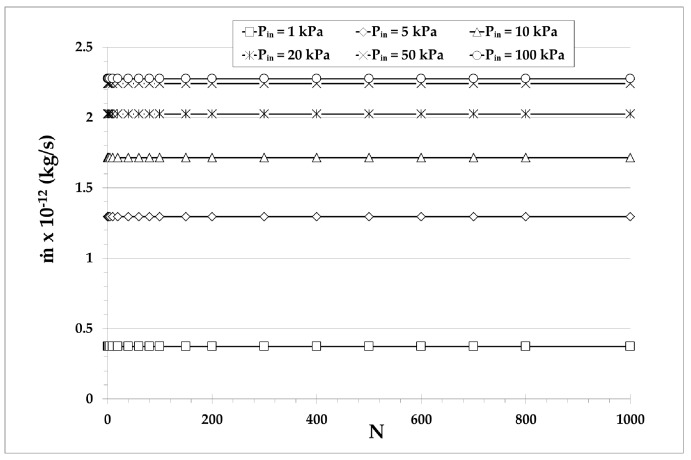
Maximum mass flow rate (corresponding to zero pressure difference) of pump B with single narrow and wide microchannels of diameters d=10 μm and D=100 μm, respectively, in each stage, at various inlet pressures, versus the number of stages.

**Figure 8 micromachines-10-00249-f008:**
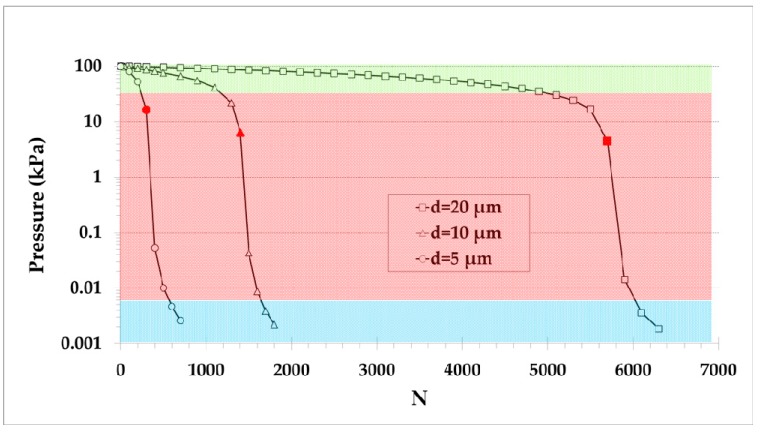
Inlet pressure evolution of a system connected to pump B as a function of its number of stages, considering narrow diameter channels d=5,10,20 μm and a constant outlet pressure $P_{out}=100$ kPa (red solid symbols represent the pressure and stage number, where the maximum slope, corresponding to the peak values of Figure 5a, is observed).

**Figure 9 micromachines-10-00249-f009:**
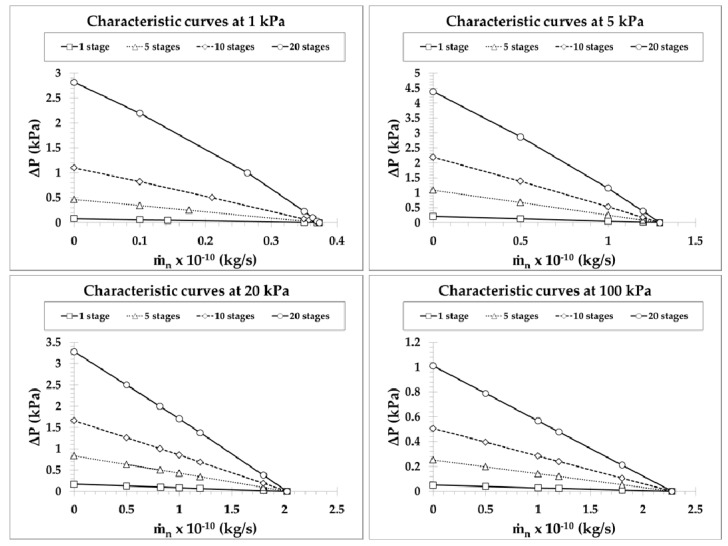
Performance characteristic curves of a Knudsen pump, based on pump C, for a number of stages N=1,5,10,20 with inlet pressures $P_{in}=1,5,20,100$ kPa when narrow microchannels diameter d=10 μm and wide channel diameter D=100 μm.

**Figure 10 micromachines-10-00249-f010:**
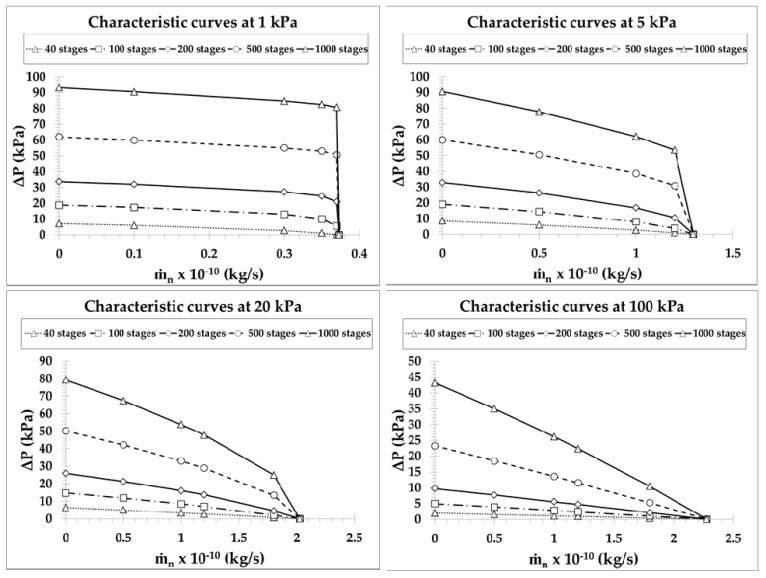
Performance characteristic curves of a Knudsen pump, based on pump C, for a number of stages N=40,100,200,500,1000 with inlet pressures $P_{in}=1,5,20,100$ kPa when narrow microchannels diameter d=10 μm and wide channel diameter D=100 μm.

**Table 1 micromachines-10-00249-t001:** Layout data in terms of diameter, number of microchannels, and elementary square area.

Total Layout Area *a × a* (μm × μm)	Microchannel Diameter *d* (μm)	Number *n* of Parallel Microchannels
200 × 200	50	4
200 × 200	20	25
200 × 200	10	100
200 × 200	5	400

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
