# Peer review of "Design Guidelines for Thermally Driven Micropumps of Different Architectures Based on Target Applications via Kinetic Modeling and Simulations"

_micromachines, 2019, doi:10.3390/mi10040249_

Round 1

Reviewer 1 Report

Please see the attached file for the review report.

Author Response

The authors would like to thank the reviewers for their in-depth analysis of the paper, which has allowed a significant improvement of its content. All comments have been taken into account and are addressed below.

A point by point response to each of the reviewer’s comments is provided in the attached document along with the updated version of the manuscript with all the changes highlighted in the text at the end of this at the end of the file.

Reviewer 2 Report

The authors theoretically investigate the performance of proposed designs for Knudsen pumps to be manufactured by lithography of dry-film photoresists. The model used is based on known solutions for net gas flow along long channels with applied thermal and pressure gradients. Three exemplary designs are thoroughly evaluated and guidelines for design and operation are drawn from these.

Knudsen compressors have attracted continuous attention both experimentally and theoretically as manufacturing capabilities of small scale structures evolve and begin to find relevant real life applications. The manuscript is well written and the presentation is clear. With this in mind, I recommend accepting the manuscript for publication in Micromachines after the minor comments below have been addressed.

Minor comments:

p.3, paragraph below figure 1: "has leaded the marked" should probably be "has been leading the market".

p.7: The maximum mass flow rate and maximum pressure difference are first introduced in the second paragraph of section 4.1, discussing pump A. Although it is plausible here that these must have been obtained for zero pressure difference and mass flow rate respectively, this is only stated explicitly while discussing pump B in section 4.2. This should already be stated in section 4.1.

p.11, figure 8. When printed without color the red symbols are hard to discern from the black ones even with the slightly thicker outline. Using full symbols for these as in figure 4 might increase the readability for this case.

p.12. While discussing the maximum pressure obtainable with Pump C it is mentioned that "these results are in agreement with the corresponding ones in figure 6". Isn't it possible to make the stronger statement that figure 6 applies unaltered to design C as well (at least for the model used)?

Kinetic coefficients G_T and G_P are listed in table A.1 for discrete values of \delta. How were the corresponding values for intermediate \delta interpolated for the simulation?

While in [9] a Knudsen compressor similar to design B of the present study is investigated assuming 2D channels, a follow-up was conducted assuming circular pipes [Aoki, 2008], as in the present study. It may be of interest to compare the results, as kinetic coefficients based on different collision models are used.

[Aoki, 2008] Aoki, Kazuo, Shigeru Takata, and Ko Kugimoto. "Diffusion approximation for the Knudsen compressor composed of circular tubes." AIP Conference Proceedings. Vol. 1084. No. 1. AIP, 2008.

Author Response

The authors would like to thank the reviewers for their in-depth analysis of the paper, which has allowed a significant improvement of its content. All comments have been taken into account and are addressed below.

A point by point response to each of the reviewer’s comments is provided in the attached document along with the updated version of the manuscript with all the changes highlighted in the text at the end of the file.

Round 2

Reviewer 1 Report

I appreciate the authors' effort to improve the manuscript. 

The revised version is fine to be published. 

In line 277, I found a typo (a chinese characte is included ) and this should be corrected before publication.